

# ISAT v2.0: An integrated tool for nested domain configurations and model-ready emission inventories for WRF-AQM

Kun Wang[1,6], Chao Gao[2], Haofan Wang[3], Kai Wu[4], Qingqing Tong[2], Mo Dan[1], Kaiyun Liu[5, *], Xiaohui Ji[1, *]

[1] Institute of urban safety and environmental science, Beijing academy of science and technology, Beijing 100054, China
[2] Key Laboratory of Wetland Ecology and Environment, Northeast Institute of Geography and Agroecology, Chinese Academy of Sciences, Changchun, 130102, China
[3] School of Atmospheric Sciences, Sun Yat-sen University, Zhuhai, Guangdong, China
[4] Department of Land, Air, and Water Resources, University of California, Davis, CA, USA
[5] State Key Joint Laboratory of Environment Simulation and Pollution Control, School of Environment, Tsinghua University, Beijing 100084, China6
[6] Key Laboratory of Marine Environmental Science and Ecology, Ministry of Education, Ocean University of China, Qingdao, 266100, China

Correspondence: Kaiyun Liu (liuky2021@tsinghua.edu.cn); Xiaohui Ji (jixiaohui@bmilp.com)

**Abstract.** An integrated tool, ISAT (Inventory Spatial Allocation Tool) v2.0, was developed to configure nested domains, downscale regional emission inventories, allocate local emission inventories, and generate model-ready emission inventories for the Weather Research and Forecasting (WRF)-Air quality numerical model (AQM). We built four modules into this tool: i) "Prepgrid" conducted a nested domain configuration algorithm based on WRF-AQM nested rules and the target domain shapefile; ii) "Downscale" established a "sub-grid nearest" method to improve both the accuracy and computational efficiency of downscaling the regional emission inventory based on spatial surrogate; iii) "Mapinv" allocated a user-defined regional/city-level emission inventory to grid level based on the target domain shapefile and the spatial surrogate; and iv) "Prepmodel" introduced unique user-friendly emission sector IDs using abbreviations and speciation profiles based on species in the emission inventory and chemical mechanisms for model-ready inventories. The inline format model-ready emission inventory built into this tool is available for both the CMAQ and CAMx. This tool will help model users establish workflow from configuring the research domain to running WRF-AQM. The tool's framework and related algorithms will also help researchers develop emission inventory processing tools for WRF-AQM.

## 1 Introduction

High spatial-temporal resolution emission inventories (HREIs) of air pollutants are essential for atmospheric environmental mitigation strategies, ambient air quality forecasting, and research on air pollution, as they can quantify pollutant concentrations based on an air quality model (AQM), such as the Weather Research and Forecast (WRF)-Community Multiscale Air Quality (CMAQ) model (Burnett et al., 2018; H. Wang et al., 2022; K. Wang et al., 2022). Preparing an HREI requires spatial allocation, temporal allocation, and chemical speciation. Spatial allocation maps areas and point emissions



into the AQM domain with source locations and spatial surrogates (Z. Huang et al., 2021; Zheng et al., 2021; Zhou et al., 2017). Hourly temporal allocation provides emissions based on product output statistics, heating degree days, hourly
variations in traffic volume, and other temporal profiles (K. Wang et al., 2021a). Chemical speciation creates speciated emissions for a specific mechanism, such as CB05 or SAPRC99, based on speciation profiles that provide the composition of organic gases and particulate matter in sectors (Huang et al., 2015).

The downscaling of regional emission inventory (REI) disaggregated atmospheric emissions from a national or regional scale to the grid level presents a method for obtaining an HREI for an AQM, such as the MEIC, with a spatial resolution of
0.25° (ECJRC, 2017; Li et al., 2017; Zheng et al., 2018). The spatial surrogate represents a fraction of the total national/regional emission on a target grid between zero and one as a downscaling process (Eyth and Hanisak, 2003). Population density, land use, and road maps were developed as proxies and spatial surrogates for the intensity of real activity in residential, agriculture, transportation, and other sectors (Lin et al., 2022; Wang et al., 2017). Nearest method is a popular downscaling method, which locates the closet REI value on the target grid and allocates emissions using spatial surrogates.
The projection in the AQM changes according to the research domain, resulting in inevitable projection differences between the AQM and the REI. The nearest method may amplify the emission mismatch without considering a projection difference. The intersect method splits the target domain into several parts between the target domain and the REI, which accounts for the projection difference but may lead to low calculation efficiency. Therefore, it was necessary to develop a new algorithm to improve both calculation efficiency and accuracy in the REI downscaling process.
Although various emission inventory processing tools have been previously developed, there is still no integrated tool that can address the domain configuration to model-ready inventory workflow for WRF-AQMs. For example, TEMMS can only support the emission of transportation (Namdeo et al., 2002), and THOSCANE cannot create a model-ready emission inventory for AQM (Monforti and Pederzoli, 2005). SMOKE, Linux-platform supported and widely used in AQM, requires a predefined spatial surrogate from other geoprocessing tools, such as ArcGIS and SA tools, and it cannot configure nested
domains (Baek and Seppanen, 2021). The WRF Domain Wizard (https://jiririchter.github.io/WRFDomainWizard/) allows the user to configure nested domains by manually delimiting research areas. However, without the consideration of a shapefile in the target area, this tool cannot obtain accurate domains and requires several trials and expert experience to obtain suitable nested domains in AQMs. Therefore, there is a need to develop an integrated tool that ensures domain consistency and provides an easy-to-use workflow, from a nested domain configuration to a WRF-AQM system.
Innovatively, we developed a user-friendly workflow from a nested domain configuration to a model-ready emission inventory for WRF-AQM. We established a shapefile-based nested domain configuration algorithm that guaranteed the consistency of the domain between the WRF model, AQM, emission inventory and domain configuration parameters for the WRF and AQM. We proposed a regional inventory downscaling algorithm named the "sub-grid nearest method," which improved the calculation efficiency and accuracy of the REI downscaling. We provided interfaces for a model-ready
emission inventory for CMAQ and CAMx models. Moreover, we embedded population- and road-based spatial surrogate



data into the ISAT v2.0, which supports most spatial allocation workflows for WRF-AQM in China. This study is expected to provide a user-friendly and integrated framework for an emission inventory processing tool for WRF-AQM.

## 2 Materials and methods

We developed an integrated tool to complete the workflow from a nested domain configuration to a model-ready HREI for
WRF-AQM, as depicted in Figure 1. The shapefile for the study area was a basic nested domain configuration file with determinate regional attributes for grids. Both gridded REIs, such as the MEIC, and local emission inventories can be processed into user-defined domains by "Downscale" and "Mapinv" modules. Moreover, the inline format model-ready emission inventories for CMAQ and CAMx models can be created in "Prepmodel" based on previous modules used in this workflow and CMAQ model-instrumented support modules, such as ISAM and DDM. Users can adopt each module
individually or in combination according to their needs. Moreover, our methodological innovations included a nested domain configuration algorithm, a sub-grid nearest method, a user-friendly emission source ID and speciation profiles, as detailed in the following sections.

### 2.1 Nested domain configuration method

Nested domains can be classified into parent or child domains according to the nesting rules. The parent domain, which is
the outermost domain, determines the projection in WRF-AQM. Using Arakawa-C grid staggering for the WRF model, 3:1 grid-distance nesting ratio is required for nested domain configurations (Skamarock et al., 2019). Grid numbers in child domains are constrained by grid-distance nesting ratios between domains. Considering the shapefile of a research area, the grid-distance nesting ratio, user-defined grid parameters, and the widely used lambert conformal conic (LCC) projection, we established a workflow for nested domain configuration. User-defined grid numbers in the east, west, south, and north
directions were increased to modify the domain by need: ($add_e, add_w, add_s$ and $add_n$). The number of grids that must be added under grid-distance nesting rules in the WRF model was also considered: ($add_x$ and $add_y$). Taking the $x$-direction as an example, $add_e, add_w$ and $add_x$ were used to define the domain in the WRF model, as shown in Figure 2. Moreover, "*model_clip*" was conduct to define lateral boundary for AQM, as shown in **Appendix. A**. The major steps and key parameters are described below.

### 2.1.1 Projection configuration based on the parent domain

Standard parallels, central meridian, and the LCC latitude of projection can be obtained from the shapefile in the parent domain. Due to defined grid spacing, the WRF-AQM domain may exceed the study area's shapefile; thus, the shapefile center does not represent the center of the LCC projection. Therefore, we designed four main steps to accomplish projection configuration: i) Standard parallels were defined by users according to the location of the parent domain (for example,
Shanxi province configured with 33° and 42°). ii) Under LCC projection in the first step and the shapefile, we obtained the





number of grids ($num_x$ and $num_y$) and origin ($X_{min}$ and $Y_{min}$) in the parent domain, as shown in Eqs. (1–4). iii) Transferred center of the domain into WGS 84 projection. iv) Looped through steps "iii) –ii) – iii)" until the difference in the central longitude and latitude between the loops could be ignored.

$$num_x = Roundup\left(\frac{disx}{dx}\right) + add_e + add_w \,,\tag{1}$$

$$num_y = Roundup\left(\frac{disy}{dy}\right) + add_s + add_n,\tag{2}$$

$$X_{min} = \frac{-(num_x \times dx)}{2},\tag{3}$$

$$Y_{min} = \frac{-(num_y \times dy)}{2},\tag{4}$$

where, $disx$ and $disy$ is the length of the shapefile in the $x$- and $y$-directions.

**2.1.2 Determination of child domain parameters**

Constrained by grid-distance nesting rules and lateral boundaries for the AQM, the child domain configuration step is different from the parent domain step. Spatial resolution and origin in the parent and child domains ($dx_{par}$, $dx_{child}$, $xstart_{child}$, and $xstart_{parent}$) were adopted for the child domain configuration. Taking the $x$-direction as an example, the number of grids was calculated based on the spatial coverage of the shapefile, user-defined parameters, and the number of grids constrained by the nesting rules in the WRF model, as shown in Eq. (5). $add_x$ is the must-add number of grids according to the nesting rules and the shapefile in the parent and child domain, as shown in Eq. (6), and $num_x$ was further corrected by Eq. (7-8). $X_{min}$ is the present start position of the child domain with a projection configuration based on the nesting rules and the user-defined $add_w$ as shown in Eq. (9). Moreover, $start_x$ denotes the starting position for the child domain in the parent domain, which can be used to configure nested domain in the WRF model in Eq. (10).

$$num_{x,tmp} = Roundup\left(\frac{disx_{child}}{dx_{child}}\right) + add_e + add_w + add_x,\tag{5}$$

$$add_x = Roundup\left(\frac{xstart_{child} - int\left(\frac{xstart_{child} - xstart_{parent}}{dx_{par}}\right) \times dx_{par} - xstart_{parent}}{dx_{child}}\right),\tag{6}$$

$$tmpgrid_x = mod\left(num_{x,tmp}, \frac{dx_{par}}{dx_{child}}\right),\tag{7}$$

$$num_x = \begin{cases} num_{x,tmp} \quad (tmpgrid_x = 0) \\ num_{x,tmp} + \frac{dx_{par}}{dx_{child}} - tmpgrid_x \quad (tmpgrid_x \neq 0) \end{cases},\tag{8}$$

$$X_{min} = int\left(\frac{xstart_{child} - xstart_{parent}}{dx_{par}}\right) \times dx_{par} + xstart_{parent} - dx_{child} * add_w,\tag{9}$$





$$start_x = \frac{X_{min} - xstart_{parent}}{dx_{par}}, \tag{10}$$

## 2.2 Sub-grid nearest method in the downscaling process

Intersect and nearest methods are commonly used to downscale REI into user-defined domains. The intersect method obtains fragments of intersection between the REI and the target domain, while the nearest method locates the nearest value from the REI. The intersect method accurately reflects the spatial relationship between the target domain and the REI. However, if there are evident projection and spatial resolution differences between REI and the target domain, a large number of small intersections are created, which significantly decreases calculation efficiency. The nearest method locates the nearest REI value for the target grid and allocates emissions based on the corresponding spatial surrogate, but it can lead to emission mismatches with close spatial resolutions and different coordinate systems. For example, the nearest method ignored the orange grid emissions but exaggerated the contribution of the yellow grids in the research domain, as shown in Figure 3. To fully consider the advantages and disadvantages of these two algorithms, we used the "sub-grid nearest" method to allocate the emission inventory, as shown in Eqs. (11–13). Figure 3 depicts the process we used to obtain the REI values using the nearest and sub-grid nearest methods. The colors in the figure represent the REI values, and the symbols (such as *) represent the grid ID. First, we introduced a "sub-grid ratio" to divide each grid into $p$ sub-grids, such as nine sub-grids in each grid. Second, we adopted the conventional nearest method to locate the nearest REI values for each sub-grid. Third, we calculated the emissions for each sub-grid based on the nearest REI value and spatial surrogates. Finally, we obtained the emissions for each grid based on a summary of the sub-grid emissions in each grid.

$$E_{target,i,j} = \sum_p E_{target,i,j,p}, \tag{11}$$

$$E_{targe,i,j,p} = E_{region,I_{i,jp}} \times \frac{SA_{target,i,j,p}}{SA_{region,I_{i,j,p}}}, \tag{12}$$

$$I_{i,j,p} = Nearest\,(Grid_{targe,i,j,p}, Grid_{region}), \tag{13}$$

where, $i,j$ are the column and row number of the target grid; $p$ is the sub-grid ID in grid $(i,j)$; $E_{target,i,j}$ is the emissions in grid $(i,j)$; $E_{target,i,j,p}$ is the emissions in sub-grid $p$ in grid $(i,j)$; $I_{i,j,p}$ is the location of sub-grid $p$ for the target grid $(i,j)$ in the region domain; $E_{region,I_{i,jp}}$ are the regional emissions in $I_{i,j,p}$; $SA_{region,I_{i,j,p}}$ is the spatial allocator for region grid $I_{i,j,p}$; $SA_{target,i,j,p}$ is the spatial allocator for sub-grid $p$ of target grid $(i,j)$; $Grid_{region}$ is the domain of the REI; and $Grid_{targe,i,j,p}$ is the sub-grid $p$ of target grid $(i,j)$.

## 2.3 User-friendly emission source sector ID

User-friendly emission source sector IDs (EIDs) are source abbreviations, for example, AG for agriculture. EID is a unique ID for the emission source sector in ISAT and defined new sources friendly. Speciate and temporal profiles for sectors can


be easily defined with EID, as shown in **Table 1**, where the speciate profile and point emission inventory are prefixed by "Speciate_" and "STACK_GROUP_" and suffixed by an EID, respectively. The area emission inventory was named EID. To define a temporal profile for each sector, users can add columns labeled by EID in hourly.csv, weekly.csv, and
monthly.csv, as shown in **Table 2**.

### 2.4 User-friendly speciation profile structure

Compared with multiple input file types for chemical speciation in the SMOKE model, including GSCNV, GSPRO, GSPRODESC, GSPRO_COMBO, GSREF, and GSTAG files, this tool represents a simple speciation process with an individual speciate profile for each sector. An example of a speciate profile is shown in **Table 3**. Here, "pollutant" denotes
the pollutants in the emission inventory; "species" denotes the pollutants in the chemical mechanism adopted in the AQM; "split_factor" represents the fraction of the "pollutant" emission to "species" in the AQM between zero and one; "unit" is the unit of species adopted in the AQM, with 'moles/s' for gaseous pollutants and 'g/s' for particulate pollutants; and "divisor" converts mass-based speciation into mole-based speciation, with molar mass for gaseous pollutants and a default value (set to 1) for particulate pollutants. Therefore, the user can easily modify the speciation profile based on the chemical mechanism
in the AQM and the pollutants in the emission inventory.

Moreover, we embedded a gridded spatial surrogate for roads and populations in the ISAT. The road database was obtained from the OpenStreetMap, with four levels, including motorway, secondary, primary, and residential road. The standard road length (Zheng et al., 2009), based on vehicle speed, number of lanes, and road width from the Code for Transport Planning on Urban Roads (GB 50220-95) in China, was used to calculate the gridded road map data. Population density data based on
the 2020 LandScan Global Population Database (Rose et al., 2021) can used in ISAT with NetCDF format. In addition, due to the nearly 1km resolution of the ISAT-embedded gridded population and road maps, users can generate HREIs with finer spatial resolution ≥1km.

### 3 Case studies

As the capital of China, Beijing has a large population and many vehicles, making it a hotspot for air pollution research in
China (Gao et al., 2018; X. Huang et al., 2021). This study presents three nested domain cases: mainland China, Beijing-Tianjin-Hebei (BTH), and Beijing (BJ), with spatial resolutions of 27, 9, and 3 km, respectively, as shown in Figure 4. The ISAT was used for the workflow from the nested domain configuration to the model-ready inventory for the WRF-AQM. The shapefile for these domains was used to configure nested domains in the WRF-AQM and to create the ready emission inventory in the "Prepgrid" module. Provincial environmental pollutant census data for the transportation and residential
sectors (BMEEP, 2020) were used to allocate local emission inventory to grid level in the "Mapinv" module. The MEIC regional emission inventory was downscaled to grid level in the "Downscale" module.



### 3.1 Results of the nested domains

ISAT provides comprehensive grid information for the WRF-AQM in comma-separated values (csv) file and shapefile formats. Csv format data contain grid information for each domain and can be further used as input for other ISAT modules. Shapefile formatted data can be displayed directly in ArcGIS or other GIS platforms to display and check the configuration for nested domains. "Domainname" and "Casename" were used to mark the output in the "Prepgrid" module. "${Casename}_gridinfo.csv" summarizes grid information in nested domains and can be used to configure domain parameters in the WRF model, as shown in Figure 5(b). "wrf_${Domainname}.shp" and "wrf_${Domainname}.csv" provided shapefile and grid information for the ISAT-configured WRF model. "aqm_${Domainname}.shp" and "aqm_${Domainname}.csv" were the shapefile and grid information for the ISAT-configured AQM model. The ISAT-configured domains were able to reflect the AQM lateral boundary demands and accurately obtain the WRF-AQM domain.

### 3.2 Downscaling of regional emissions inventory

The MEIC inventory is widely used in China. In this study, the 2020 annual MEIC emissions were preprocessed into csv format and downscaled into the AQM domain. Gridded residential emissions in BTH with a resolution of 9 km were compared with the intersect using the nearest and sub-grid nearest methods. The sub-grid ratio, the key parameter in the sub-grid nearest method, must be either one or a multiple of three. In this study, a different sub-grid ratio was applied to BTH to test the allocation result. The nearest method can be configured with a sub-grid ratio equal to one. With the sub-grid ratio increased from one to nine, the spatial allocation was close to the result of the intersect method for the urban area, as shown in Figure 6. The sub-grid method with a sub-grid ratio equal to three significantly improved the allocation compared to the nearest method, and the R squared increased from 0.91 to 0.98. The increase in the sub-grid ratio from three to nine slightly improved the distribution results, with R squared increasing from 0.979 to 0.984, as shown in Figure 7. Although a higher sub-grid ratio may lead to a more accurate allocation, one needs to consider the computational efficiency and resolution of the spatial surrogate. In addition, for REI grids with empty spatial surrogates, ISAT applies the grid area as a weight when allocating emissions. Constrained by the spatial surrogate ($\leq$1) for each REI grid, the downscaled emission inventory exhibited good agreement with the REI. It should be noted that there was some uncertainty in the boundary due to some REI grids exceeding the domain, making it difficult to constrain emissions.

### 3.3 Mapping the local emission inventory

Differing from the downscale process used for the gridded REI described in Section 3.2, mapping the provincial/city-level local emission inventory onto the grid level required the identification of region attributes for each AQM domain grid and the allocation of emissions based on spatial surrogates. In ISAT v2.0, user can add a column to define the region in the shapefile and place the local emission inventory in the "localinv" file. ISAT will allocate each species in the local emission inventory to grid level. Taking Beijing as an example, emissions for the occupied residential (RE) sector were 7,763 t NOx



and 25,900 t VOCs, and 22,700 t NOx and 2,654 t VOCs for the transportation (TR) sector. The gridded emission inventory for the RE and TR sectors created by the ISAT reflected the spatial characteristics of the population density and the road map in Beijing, as shown in Figure 8. Based on the embedded population density and road map, ISAT could flexibly allocate the local emission inventory to grid level.

## 3.4 Generating model-ready emission inventory for the AQM

User can generate inline format model-ready emission inventories for AQM using "Prepmodel" module. The gridded emission inventory, generated by "Mapinv" and "Downscale," can be used as area emission input files. The point emission file was organized into csv format with emission, locations, and stack parameters with reference to the SMOKE model. Temporal profile files were defined by statistical data and from previous studies. The speciation profiles for sectors were obtained from either the SPECIATE database or field measurements. Moreover, GRIDCRO2D files generated by the Meteorology-Chemistry Interface Processor in the CMAQ model was applied to provide global attributes and configurations consistent with the AQM. "Runtime" was used to define the model-ready emission timesteps for the AQM. The ISAT inline format emission inventory was named EID. For example, "TR.nc" denoted the area emission source for the TR sector. "PPpoint.nc" and "STACK_GROUP_PP.nc" represented hourly emission and the source information for the power plants, respectively. User can use this model-ready emission inventory directly in AQMs, such as CMAQ and CAMx, and can further apply it to instrumental modules, such as DDM and ISAM. Our previous studies used this module and achieved reliable simulations result (H. Wang et al., 2021; K. Wang et al., 2022, 2021a, 2021b). The configurations and steps used in these cases are detailed in the Appendix. A.

## 4 Conclusion

We developed an integrated tool, ISAT v2.0, to accomplish the workflow from nested domain configuration to model-ready emission inventory for WRF-AQM. The "Prepgrid" module can accurately provide user-defined domain configuration under nesting rules in WRF-AQM and the shapefile of the study area. The domain parameters in "Prepgrid" can also be used to configure the namelist.wps file in WPS. The "Downscale" module conducted with the "sub-grid nearest" method significantly improved the allocation result with the nearest method, and it allows users to flexibly set the "sub-grid ratio" according to their needs. The "Mapinv" module provides a user-friendly tool to allocate provincial/city-level local emission inventory to grid level using user-defined shapefiles and local emission inventories. The "Prepmodel" can generate an inline format model-ready file emission inventory for WRF-AQM and carry out numerical simulations for users. ISAT is free, expansible, and supported by Linux and Windows. It is easy to apply ISAT to other regional emission inventories, such as REAS, by providing grid templates and updating user-defined spatial surrogates. A benefit of the established relationship between the shapefile and the WRF-AQM domain, ISAT is highly extendable, including the creation of oceanfiles for CMAQ models, the labeling of source regions in CAMX-PSAT, and other potential uses.



*Code and data availability.* The current version of ISAT is available from the project website: https://github.com/ISAT-Office/ISAT under the GNU General Public License v3. The source code of ISAT v2.0 and input data used to produce the results used in this paper is archived on Zenodo at https://doi.org/10.5281/zenodo.7481439 (Wang *et al.,* 2022).

*Author contribution.* Kun Wang: Conceptualization, Methodology, Software, Writing-Original draft. Chao Gao, Haofan Wang and Kai Wu: Data Curation; Mo Dan: Writing-Reviewing and Editing; Kaiyun Liu and Xiaohui Ji: Resources. Supervision, Investigation.

*Competing interests.* No competing interests are present.

*Acknowledgements.* This work was supported by the National Key Research and Development Program of China (2019YFE0194500), Natural Science Foundation of China Youth Project (42207248), BJAST Scientific Research in 2022(11000022T000000468154), BJAST Scientific Research in 2022(11000022T000000468176), and the European Union's Horizon 2020    Research and Innovation programmes under grant agreement no 870301 (AQ-WATCH).

## Appendix A. ISAT step-by-step guide

Taking nested domains in "Mainland China-BTH-BJ" with spatial resolution of "27km-9km-3km" as example.

**Step 1: Obtain nested domain parameters for WRF-AQM based on domain shapefile.**

(a)  Give the information of "par.ini" in "Prepgrid" model.

| | |
|---|---|
| lat1:33.0 | # First standard latitude for LCC |
| lat2:42.0 | # Second standard latitude for LCC |
| casename:3nestdomain | # Case name |
| numdom:3 | # Number of nested domains |
| shpath: ./shp/mainlandchina.shp, ./shp/JJJ.shp,./shp/beijing.shp | # Shpfile path for domains |
| dx:27000,9000,3000 | # Resolution in meters for domains |
| xladd:2,3,3 | #Added grid in left in x direction |
| xradd:2,3,3 | #Added grid in right in x direction |
| ytadd:2,3,3 | #Added grid in top in y direction |
| ydadd:2,3,3 | #Added grid in down in y direction |
| domainname:MainlandChina,JJJ,Beijing | #Name for domains |
| model_clip:1,1,1 | #Clip number of grids for AQM |

(b)  Running prepgrid module by "python prepgrid.py". Domain information summarized in "3nestdomain_gridinfo.csv", detailed below.

| domid | centlon | centlat | xmin | ymin | numx | numy | xstart | ystart |
|---|---|---|---|---|---|---|---|---|





| 0 | 102.0737 | 36.68731 | -2484000 | -2119500 | 184 | 157 | 0 | 0 |
| 1 | 102.0737 | 36.68731 | 945000 | -40500 | 66 | 93 | 127 | 77 |
| 2 | 102.0737 | 36.68731 | 1116000 | 382500 | 63 | 72 | 19 | 47 |

Therein, you can define WPS domain using variables above:

| parent_id | 1 | 1 | 2 |
|---|---|---|---|
| parent_grid_ratio | 1 | 3 | 3 |
| i_parent_start | 1 | $xstart(1)+1 | $xstart(2)+1 |
| j_parent_start | 1 | $ystart(1)+1 | $ystart(2)+1 |
| e_we | $numx(0)+1 | $numx(1)+1 | $numx(2)+1 |
| e_sn | $numy(0)+1 | $numy(1)+1 | $numy(2)+1 |
| ref_lat | $centlat | | |
| ref_lon | $centlon | | |
| truelat1 | $lat1 | | |
| stand_lon | $centlon | | |
| truelat2 | $lat2 | | |

(c) You can get fishnet shapefile for WRF model named "wrf_"+${domainname}+ ".shp" and AQM model named "aqm_" +$domainname+ ".shp".

(d) You can also get gridded information for domains in "wrf_" +${domainname}+ ".csv" for WRF model and "aqm_" +$domainname+ ".shp" for AQM model respectively.

**Step 2: Downscaling regional emission inventory into target domains.**

(a) Give the information of "par.ini" in "Downscale" module.

| invf:./input/domain/aqmJJJ.csv | # Path of target domain information |
|---|---|
| dx:9000 | # Resolution in meters for target domain |
| ratio:3 | # Subgrid ratio |
| emissions: ./regioninv/agricultureannual.csv,./regioninv/transportationannual.csv,./regioninv/residentialannual.csv | # Path for regional emission inventory |
| casename:JJJ | # Case name |
| method:area,road,pop | # Allocation method for specific emission inventory<br>area: based on grid area<br>road: based on roads map<br>pop: based on population map |





(b) Running downscale module by "python downscale.py". And getting gridded emission for specific area sources.

**Step 3: Preparation of model-ready emission inventories for AQM.**

(a) Give the information of "par.ini" in "prepmodel" module.

| | |
|---|---|
| runtime:193 | # Time steps for AQM run |
| gridcro2d: ./src/met/GRIDCRO2D_9km | #MCIP generated GRIDCRO2D file |
| speciate: ./src/speciate/speciate_AR.csv, ./src/speciate/speciate_AG.csv | # Speciation profile for area emission sources. |
| speciate_groups: ./src/speciate/speciate_IN.csv | # Speciation profile for point emission sources. |
| temporary_hour : ./src/temporary/hourly.csv | # Hourly temporal profile for emission sources. |
| temporary_week : ./src/temporary/weekly.csv | # Weekly temporal profile for emission sources. |
| temporary_month: ./src/temporary/monthly.csv | # Monthly temporal profile for emission sources. |
| emissions: ./src/emissions/MEIC/AR.csv, ./src/emissions/MEIC/AG.csv | #Path of area emission files |
| stack_groups: ./src/emissions/MEIC/STACK_GROUP_IN.csv | #Path of point emission files |

(b) Run prepmodel module by "python area_emis.py; python point_emis.py; python point_info.py". For CAMx model, CMAQ2CAMx can be used transfer this output into CAMx.

**Other step: Mapping local emission inventory into grid level.**

(a) Give the information of "par.ini" in "mapinv" model.

| | |
|---|---|
| regionshp:./input/Region/BJ.shp | # Shapefile for region |
| Key:NAME | #Region column in shapefile |
| grid:./input/Grid/3km.csv | # Grid information by "pregrid' |
| sapath:./input/SA/inv_sa_tmp_road.csv,./input/SA/inv_sa_tmp_pop.csv | # Spatil surrogated by "downscale" module |
| localinv:./input/localinv/TR.csv,./input/localinv/RE.csv | # Path way for local emission inventory |
| localname:TR3KM,AR3KM | # Name of gridded emission inventory output |

(b) Run mapinv module by "python mapinv.py"

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





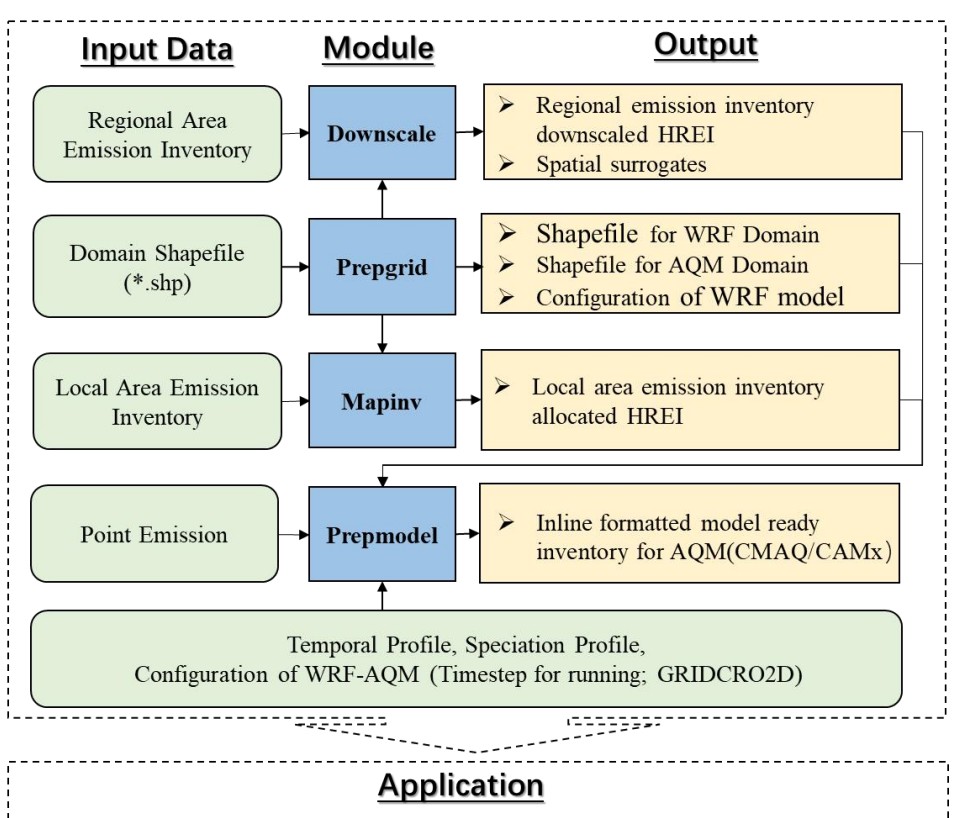

**Figure 1. ISAT framework.**

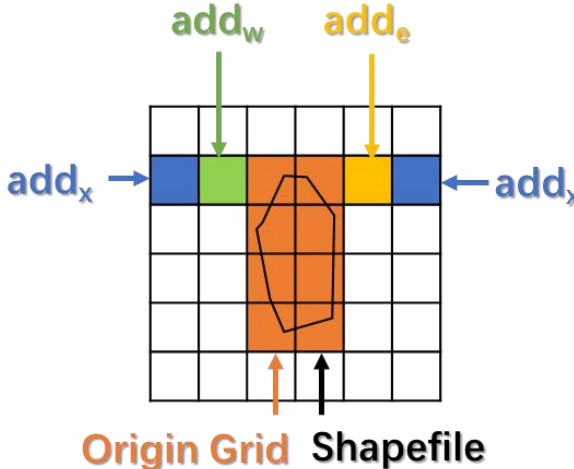


**Figure 2. Parameters for modifying the domain in the x-direction.**





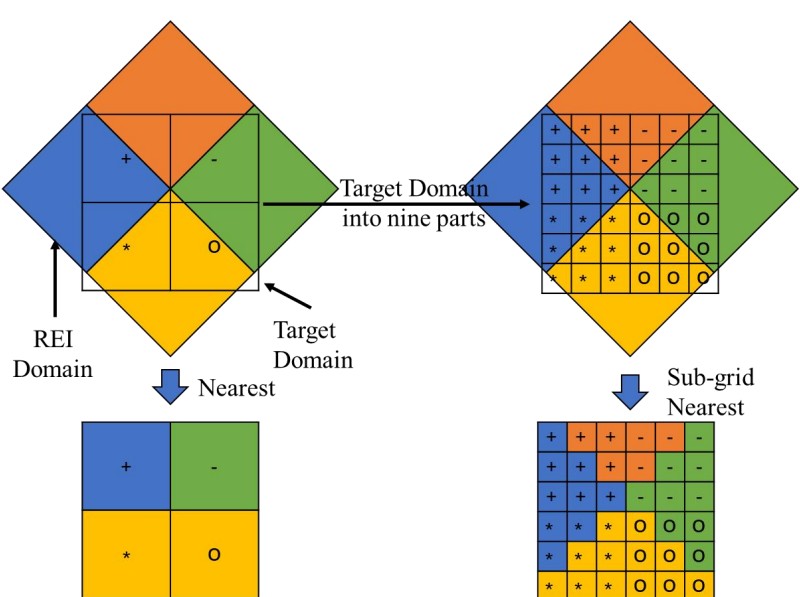

"+", "-", "*" and "o" denote grid ID in target domain

**Figure 3. Concept of the sub-grid nearest method.**

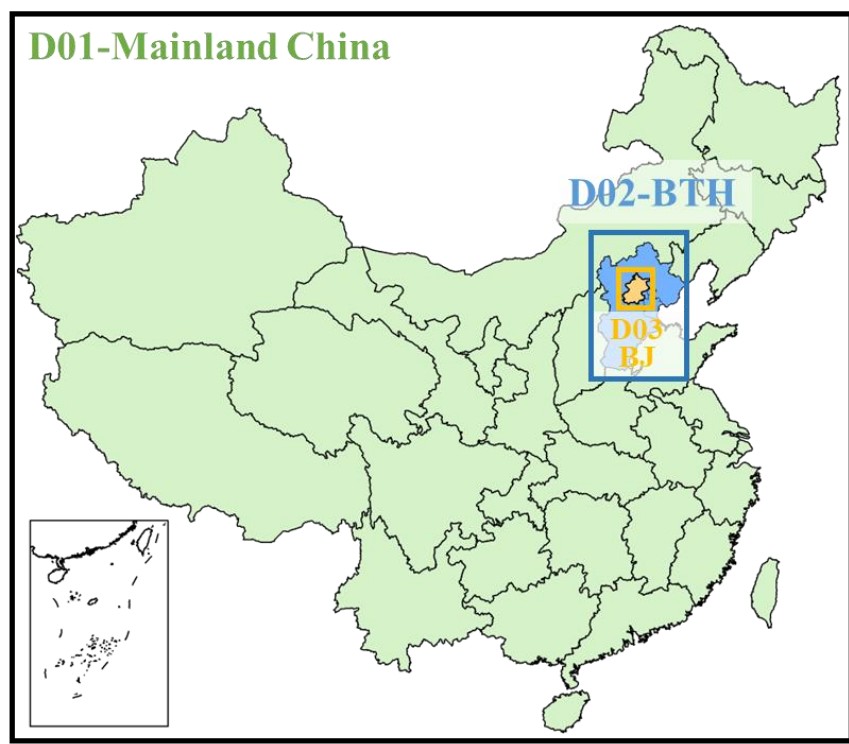

**Figure 4. The three nested domains and their shapefiles.**




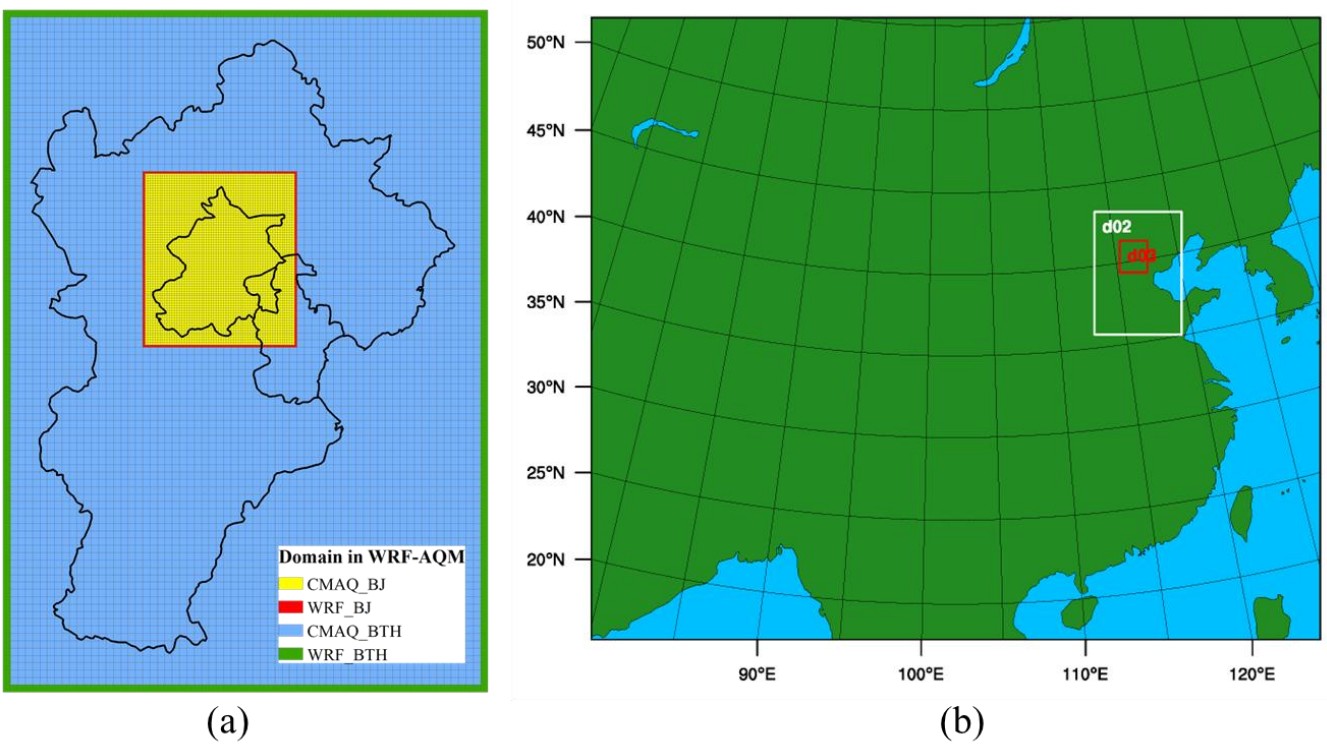

**Figure 5. ISAT-configured WRF-AQM nested domains.**

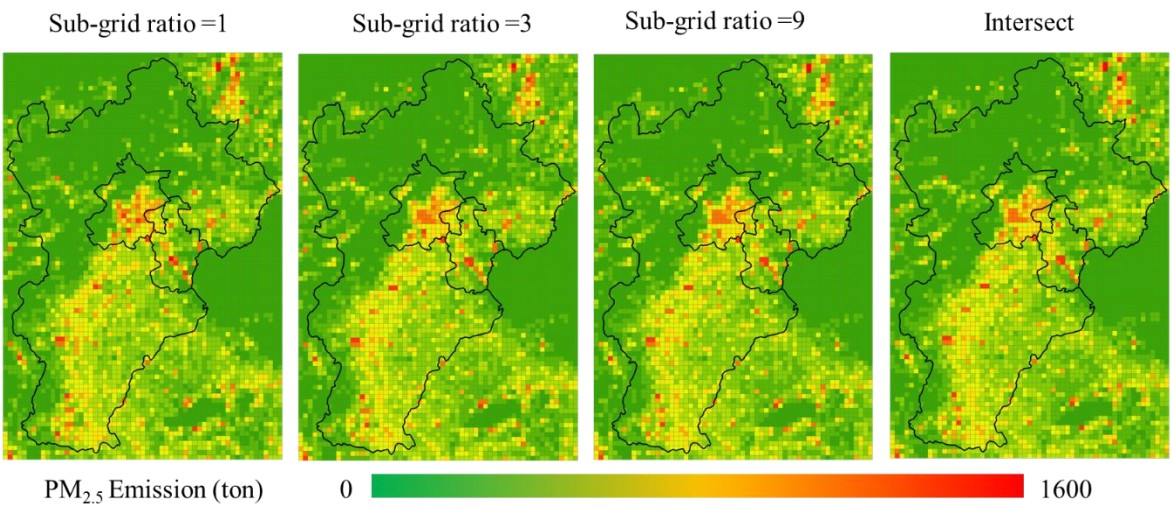

**Figure 6. Comparison of the intersect and the sub-grid nearest method allocations with different sub-grid ratios.**



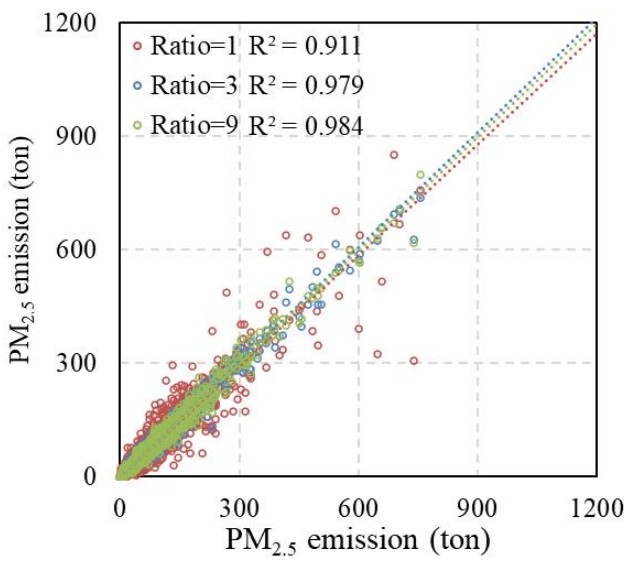

**Figure 7. Pairwise relationship between the intersect and sub-grid nearest methods with different sub-grid ratios for particulate matter (PM).**

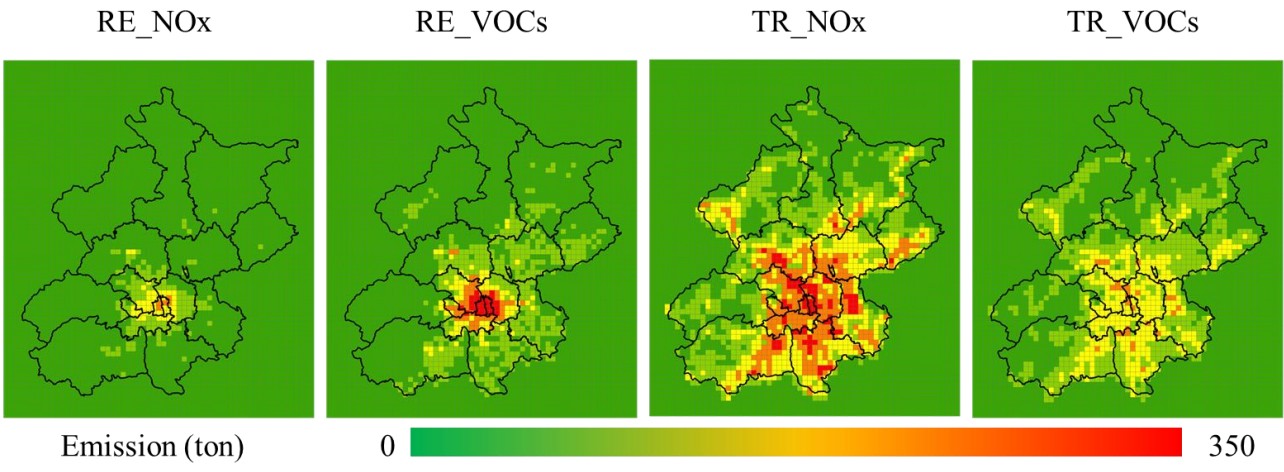


**Figure 8. Spatial allocation for Beijing.**





**Table 1. File structure defined by emission source sector ID (EID)**

| File Type | Filename | Example |
|---|---|---|
| Speciate profile | Speicate_$EID.csv | Speciate_IN.csv;   Speciate_PP.csv;   Speciate_AG.csv; Speciate_AR.csv |
| Area Emission | $EID.csv | AR.csv; AG.csv |
| Point Emission | STACK_GROUP_$EID.csv | STACK_GROUP_IN.csv; STACK_GROUP_PP.csv |

**Table 2. Case of temporal profiles (for PP source)**

| hourly.csv | | weekly.csv | | monthly.csv | |
|---|---|---|---|---|---|
| hourly* | PP | weekly | PP | monthly | PP |
| 0 | 0.045 | 0 | 0.145 | 1 | 0.090 |
| 1 | 0.047 | 1 | 0.144 | 2 | 0.065 |
| 2 | 0.046 | 2 | 0.143 | 3 | 0.083 |
| 3 | 0.045 | 3 | 0.144 | 4 | 0.075 |
| 4 | 0.043 | 4 | 0.143 | 5 | 0.075 |
| … | … | 5 | 0.138 | … | … |
| 23 | 0.044 | 6 | 0.143 | 12 | 0.098 |

*: UTC time

**Table 3. Case of speciate profile in ISAT**

| pollutant | species | split_factor | divisor | unit | pollutant | species | split_factor | divisor | unit |
|---|---|---|---|---|---|---|---|---|---|
| CB05_ALD2 | ALD2 | 1.00 | 44 | mole/s | PM2.5 | PH2O | 0.00 | 1 | g/s |
| CB05_ALDX | ALDX | 1.00 | 45 | mole/s | PM2.5 | PK | 0.00 | 1 | g/s |
| CB05_ETH | ETH | 1.00 | 28 | mole/s | PM2.5 | PMG | 0.00 | 1 | g/s |
| CB05_ETHA | ETHA | 1.00 | 30 | mole/s | PM2.5 | PMN | 0.00 | 1 | g/s |
| CB05_ETOH | ETOH | 1.00 | 46 | mole/s | PM2.5 | PMOTHR | 0.06 | 1 | g/s |
| CB05_FORM | FORM | 1.00 | 30 | mole/s | PM2.5 | PNA | 0.00 | 1 | g/s |
| CB05_IOLE | IOLE | 1.00 | 55 | mole/s | PM2.5 | PNCOM | 0.10 | 1 | g/s |
| CB05_ISOP | ISOP | 1.00 | 68 | mole/s | PM2.5 | PNH4 | 0.01 | 1 | g/s |
| CB05_MEOH | MEOH | 1.00 | 32 | mole/s | PM2.5 | PNO3 | 0.00 | 1 | g/s |
| CB05_NVOL | NVOL | 1.00 | 1 | mole/s | PM2.5 | POC | 0.63 | 1 | g/s |
| CB05_OLE | OLE | 1.00 | 30 | mole/s | PM2.5 | PSI | 0.00 | 1 | g/s |
| CB05_PAR | PAR | 1.00 | 17 | mole/s | PM2.5 | PSO4 | 0.01 | 1 | g/s |
| CB05_TERP | TERP | 1.00 | 136 | mole/s | PM2.5 | PTI | 0.00 | 1 | g/s |
| CB05_TOL | TOL | 1.00 | 101 | mole/s | SO2 | SO2 | 1.00 | 64 | mole/s |
| CB05_UNR | UNR | 1.00 | 19 | mole/s | SO2 | SULF | 0.01 | 98 | mole/s |
| CB05_XYL | XYL | 1.00 | 114 | mole/s | NH3 | NH3 | 1.00 | 17 | mole/s |
| PM2.5 | PAL | 0.00 | 1 | g/s | CO | CO | 1.00 | 28 | mole/s |
| PM2.5 | PCA | 0.00 | 1 | g/s | PMcoarse | PMC | 1.00 | 1 | g/s |
| PM2.5 | PCL | 0.00 | 1 | g/s | NOx | NO | 0.90 | 30 | mole/s |
| PM2.5 | PEC | 0.18 | 1 | g/s | NOx | NO2 | 0.09 | 46 | mole/s |
| PM2.5 | PFE | 0.00 | 1 | g/s | | | | | |