# Peer review of "ISAT v2.0: An integrated tool for nested domain configurations and model-ready emission inventories for WRF-AQM"

_Geoscientific Model Development, 2022_

## Referee Comment (RC1)

This paper describes a tool called ISAT (Inventory Spatial Allocation Tool) v2.0, that was developed to configure nested domains, downscale regional emission inventories, allocate local emission inventories, and generate model-ready emission inventories for the Weather Research and Forecasting (WRF) coupled with Air Quality models. The model is described and freely accessible on a github repository. An application is proposed over a region of China.

My general impression is that the paper deserves to be published with some adjustments in the description of the steps in the core manuscript. I appreciate the fact that most of details are reported in appendices but probably more explanations are needed in the core of the paper. Sometimes the description is a bit cryptic or too elliptic. In general the English is fair and correct, but some part needs to be rephrased.

**Major comments**

My first major comment is to reshape the presentation to better identify the 4 steps in the abstract after in the publications in the subsections.

My second major comment is to show that this pre-processing is mass conservative, some numbers and a section dedicated to the test of mass conservation are needed. It can be short, but the author must inform us about the quality of the downscaling in terms of mass fluxes conservation.

Third, the downscaling methods should be compared with an independent bottom-up emission inventory. I know that fine scale emissions database exist over the BTH, it will be relevant to compare this methodology with a reference expected to be the closest as possible of the true. It seems the authors compare two different downscaling methods which is not relevant in my opinion.

At last the authors must inform us about the type of proxy that are adequate with precursor pollutants. For instance, for ammonia emissions, crops or other spatial proxy will be more appropriated. A table summarizing the proxy for each main precursor should be added in the core of the paper.

**Minor comments**

L31 to L36: The sentences are not correctly written, please rephrase. Some words are missing!

L43: please provide a reference here for the "nearest method"

L60-66: Rephrase this part to describe the content of the paper. The last sentence is not necessary as it is already said before.

L70-71: I do not understand this statement: "The shapefile for the study area was a basic nested domain configuration file with determinate regional attributes for grids".

L72: what do the authors mean by "inline"?

L79 : which nested rules ? be more specific

L87 : the concept of add_* is a bit difficult to understand, please clarify

L91 : what LCC means ? In general review all acronyms to make sure they are defined

L94-98 : Please use bullet points to clarify the various steps.

L117 : I my opinion equation 8 can be simplified, the second line should cover the case in the first line.

L118 : *int* is the « integer part » function, please mention it

L174 : what do the authors mean by « census » here ?

L180 : Please provide the meaning of GIS.

L185-186 : the last sentence needs to be reformulated, it is difficult to understand the wording « demand »

L188 : what is the resolution of MEIC here ?

L189 : if I understand well, the authors compare MEIC downscaled by their routine with an independent BTH emission inventory ? right ? It is not clear is this section what the authors are comparing.

L200 : what do the authors mean by «uncertainty in the boundary » ?

L207 : could the authors extend the analysis on other precursors emissions like NH3 that have a complete differrent spatial pattern ?

L237 : what are « oceanfiles » ?

---

## Author Comment (AC1)

**Response to comments of the anonymous Referee #1, "RC1"**

This paper describes a tool called ISAT (Inventory Spatial Allocation Tool) v2.0, that was developed to configure nested domains, downscale regional emission inventories, allocate local emission inventories, and generate model-ready emission inventories for the Weather Research and Forecasting (WRF) coupled with Air Quality models. The model is described and freely accessible on a GitHub repository. An application is proposed over a region of China. My general impression is that the paper deserves to be published with some adjustments in the description of the steps in the core manuscript. I appreciate the fact that most of details are reported in appendices but probably more explanations are needed in the core of the paper. Sometimes the description is a bit cryptic or too elliptic. In general, the English is fair and correct, but some part needs to be phrased.

**Author's Response:**

Thanks for your effort on reviewing our paper and valuable comments, which helped us to improve this paper.

**Detailed comments**

1. My first major comment is to reshape the presentation to better identify the 4 steps in the abstract after in the publications in the subsections.

**Author's Response:**

Thanks for this valuable comment. In ISAT v2.0, we built four independent modules to help users prepare nested domain configuration and model-ready emission inventories for WRF-AQM. To better identify four modules in main text, we modified the title of subsections as follow:

- "2.1 Nested domain configuration in "Prepgrid"
- 2.2 Sub-grid nearest method in "Downscale"
- 2.3 User-friendly emission source sector ID in "Prepmodel"

2.4 User-friendly speciation profile structure in "Prepmodel"

3.1 Nested domain configuration by "Prepgrid"

3.2 Downscaled regional emissions inventory by "Downscale"

3.3 Gridded local emission inventory by "Mapinv"

3.4 Model-ready emission inventory by "Prepmodel""

2. My second major comment is to show that this pre-processing is mass conservative, some numbers and a section dedicated to the test of mass conservation are needed. It can be short, but the author must inform us about the quality of the downscaling in terms of mass fluxes conservation.

**Author's Response:**

Thanks for this valuable comment. Mass conservation is the basic principle for emission inventory preprocessing. In revision, we added relevant description on "2 materials and method" and "3 Case studies":

(1) In "2.2 Sub-grid nearest method in "Downscale":

We downscale the regional emission inventory into target grid based on the principle of mass conservation by ratio of  $\frac{SA_{target,i,j,p}}{SA_{region,I_{i,j,p}}}$ , please see Eq (12).  $SA_{region,I_{i,j,p}}$  should be larger or equal to the summary of  $SA_{target,i,j,p}$  in same region grid  $I_{i,j,p}$ , in order to ensure mass conservation. In revision, we add a description of this constraint condition.

Please see "It should be ensured that  $SA_{region,I_{i,j,p}}$  is larger or equal to the sum of  $SA_{target,i,j,p}$ in the same region grid  $I_{i,j,p}$  based on the principle of mass conservation." on Line 135.

(2) In "2.4 User-friendly speciation profile structure in "Prepmodel"

Mass conservation is also important to speciation allocation. Please see "split\_factor" represents the fraction of the "pollutant" emission to "species" in the AQM between zero and one, and it must be mass conservative" on Line 159.

(3) In "3.2 Downscaled regional emissions inventory by "Downscale"

We evaluated the quality of the downscaling in terms of mass fluxes conservation.

Please see "Constrained by the spatial surrogate ( $\leq 1$ ) for each REI grid based on principle of mass conservation, the downscaled emission inventory exhibited good agreement with the REI. Regarding the result of intersect method as the true value, the relative error of total PM2.5 emissions under each sub-grid ratio is less than 0.2 %, which reflects the good quality of the downscaling in terms of mass flux conservation." on Line 207.

3. Third, the downscaling methods should be compared with an independent bottom-up emission inventory. I know that fine scale emissions database exist over the BTH, it will be relevant to compare this methodology with a reference expected to be the closest as possible of the true. It seems the authors compare two different downscaling methods which is not relevant in my opinion.

**Author's Response:**

Thanks for this comment.

In ISAT, the main purpose for the downscaling regional emission inventory is to provide model-ready emission inventory for outer-domain in WRF-AQM. The simulation results obtained from WRF-AQM in the outer domain provide the boundary condition for the inner domain. Therefore, the accuracy of downscaled emission inventory plays a critical role in the simulation results. However, the downscaling approach, which maps regional emission inventory into user-defined domain with spatial proxy, cannot be comparable to the bottom-up local emission inventory. For instance, the 0.1x0.1° emission inventory for the BTH region in 2017

(http://meicmodel.org.cn/) that you mentioned benefits from environmental statistics, pollution source census data, and over 8,000 industrial point sources in the region, resulting in a more detailed and realistic spatial emission characteristics than spatial proxies, such as Landscan data used in this study.

To develop an efficient downscale module in ISAT v2.0, we downscaled  $0.25 \times 0.25^{\circ}$  residential PM2.5 emission in MEIC in 2020 into 9 km×9 km domain in BTH region. Subsequently, we compared the results generated under various downscaling methods. As existed significant variance in emissions data between the years of 2017 and 2020, and between downscaled method and bottom-up method, we focus on the primary objective of our study, and we refrained comparing our downscaling results with the  $0.1 \times 0.1^{\circ}$  emission inventory in BTH.

We agree with the reviewers on the advantages of fine-scale inventories and sorry for our unclear description of the research's purpose. In the revision, we supplied a detailed explanation of the advantages of local emission inventories and made modifications to this section as follows:

(1) In "2 Materials and methods"

We modified the description of ISAT framework. Please see "For domains without local emission inventory, REIs, such as the MEIC, can be downscaled into domains by Downscale" module." on Line 71.

(2) In "3.2 Downscaled regional emissions inventory by "Downscale""

"These downscaled REI can be used to produce model-ready emission for domains without local emission inventory. Due to more detailed and realistic emission characteristics, local emission inventory is highly recommended on spatial-temporal emission characteristic research and simulation in WRF-AQM in the innermost domain." on Line 211.

4. At last the authors must inform us about the type of proxy that are adequate with precursor pollutants. For instance, for ammonia emissions, crops or other spatial proxy will be more

appropriated. A table summarizing the proxy for each main precursor should be added in the core of the paper.

**Author's Response:**

Thanks for this comment. In this study, we have established a user-friendly and integrated framework on preprocessing emission inventory. User can prepare any proxy suitable for their need in this tool. Thanks for your suggestion, we supplied some suggestions on spatial proxy for different emission sectors.

Please see "Users can also prepare any proxy for their needs. For example, land use and POIs data are recommended for agriculture and residential sectors (Wang et al., 2017; Huang et al., 2022)." on Line 168. And one reference was added.

- Huang, C., Zhuang, Q., Meng, X. et al. 2022. A fine spatial resolution modelling of urban carbon emissions: a case study of Shanghai, China. Sci Rep 12, 9255. https://doi.org/10.1038/s41598-022-13487-5
- 5. L31 to L36: The sentences are not correctly written, please rephrase. Some words are missing!

**Author's Response:**

Thanks for this comment. We rephrased this sentence. Please see "Temporal allocation provides hourly emissions based on temporal profiles, such as monthly production statistics, heating degree days and temporal variation in traffic activity" on Line 35.

**6. L43: please provide a reference here for the "nearest method"**

**Author's Response:**

Thanks for this comment. We supplied two references for "nearest method".

"Nearest method is a popular downscaling method, which locates the closet REI value on the target grid and allocates emissions using spatial surrogates (Zhang et al., 2014; Zhuang et al., 2022)." on Line 44.

- Zhang, X., Gurney, K. R., Rayner, P., et al. 2014. Sensitivity of simulated CO2 concentration to regridding of global fossil fuel CO2 emissions. Geosci. Model Dev., 7, 2867–2874, https://doi.org/10.5194/gmd-7-2867-2014.
- Zhuang, J., Dussin, R., Huard, D., et al. 2022. xESMF: v0.7.0 (v0.7.0). Zenodo. https://doi.org/10.5281/zenodo.7447707
- 7. L60-66: Rephrase this part to describe the content of the paper. The last sentence is not necessary as it is already said before.

**Author's Response:**

Thanks for this comment. We have revised the description of the content of our paper. Please see "Innovatively, a user-friendly and integrated tool, ISAT (Inventory Spatial Allocation Tool) v2.0, was developed from nested domain configuration to model-ready emission inventory for WRF-AQM in this study. We established a nested domain configuration algorithm based on shapefile that ensure domain consistency between the WRF, AQM and emission inventory. We proposed a regional inventory downscaling algorithm called the "sub-grid nearest method," which improved the computational efficiency and accuracy of the REI downscaling. We provided interfaces for a model-ready emission inventory for CMAQ and CAMx models. In addition, we embedded population- and road-based spatial surrogate data into the ISAT, which supports most spatial allocation workflows for WRF-AQM in China." on Line 61.

8. L70-71: I do not understand this statement: "The shapefile for the study area was a basic nested domain configuration file with determinate regional attributes for grids".

**Author's Response:**

We rephased this sentence. "The shapefiles provide location and extent in each domain and used to configure nested domains in WRF-AQM by "Prepgrid" module" on Line 70.

**9. L72: what do the authors mean by "inline"?**

**Author's Response:**

Both of CMAQ and CAMx model support in-line plume rise calculations and inline format model ready emission inventory. And this inline format model ready emission inventory can further apply it to CMAQ's instrumental modules, such as DDM and ISAM.

In revision, we supplied the description of inline format emission and a reference in this study. Please see "Both of CMAQ and CAMx model support inline plume rise calculations and inline format HREI (Guevara et al., 2014). Based on gridded emissions from "Downscale" and "Mapinv", the inline format HREI for WRF-AQM can be created in "Prepmodel" on Line 73.

Guevara M, Soret A, Arévalo G, et al. 2014. Implementation of plume rise and its impacts on emissions and air quality modelling [J]. Atmos. Environ., 99: 618-29.

10. L79 : which nested rules ? be more specific

**Author's Response:**

This statement was rephased, and one reference was supplied. Please see "Nested domains can be classified into parent or child domains (Daniels et al., 2016)" on Line 80.

Daniels, M. H., Lundquist, K. A., Mirocha, J. D., et al. 2016. A New Vertical Grid Nesting Capability in the Weather Research and Forecasting (WRF) Model, Monthly Weather Review, 144(10), 3725-3747.

**11. L87 : the concept of add \* is a bit difficult to understand, please clarify**

**Author's Response:**

The domains in WRF-AQM were initially configured based on the extent of shapefiles. Considering user's needs and the 3:1 grid-distance nesting ratio, the domain may need to be extended. Therefore, we use "add\_\*" denote the expend grids in the initial domain. In the revision, we rephased the description of "add\_\*".

Please see "Using Arakawa-C grid staggering in WRF model, 3:1 grid-distance nesting ratio between domains is required for nested domain configuration (Skamarock et al., 2019). Nested domains can be classified into parent and child domains and initially configured based on the extent of shapefiles (Daniels et al., 2016). The outermost domain determines the projection in WRF-AQM. The child domains need to be extended based on user needs and 3:1 grid-distance nesting ratio between domains. Users can set the number of extended grids in the east, west, south, and north directions as needed:  $(add_e, add_w, add_s and add_n)$ . The number of grids that must be expanded under grid-distance nesting rules were also considered:  $(add_x and$  $<math>add_y)$ . Taking the x-direction as an example,  $add_e, add_w$  and  $add_x$  were used to extent the domain in the WRF model, as shown in Figure 2." on Line 79.

**12. L91 : what LCC means ? In general review all acronyms to make sure they are defined**

**Author's Response:**

We supplied the full name of abbreviations of LCC. Please see "Standard parallels, central meridian, and latitude of the lambert conformal conic (LCC) projection can be obtained from the shapefile in the outmost domain." on Line 91.

**13. L94-98 : Please use bullet points to clarify the various steps.**

**Author's Response:**

We rephased this section. Please see "We designed four main steps to accomplish projection configuration: i) Setting standard parallels. Standard parallels were defined by users according to the location of the outmost domain (for example, Shanxi province configured with 33° and 42°). ii) Setting extent and number of grids in domain. Under the LCC projection in the previous step and the shapefile, we obtained the number of grids (*numx* and *numy*) and origin ( $X_{min}$  and  $Y_{min}$ ) as shown in Eqs. (1–4). iii) Updating LCC projection. Transferred center of the domain into WGS 84 projection and used to update the center in LCC projection. iv) Getting final LCC projection parameters. Looped through steps "iii) –ii) – iii)" until the difference in the center in domain and shapefile in the loops could be ignored." on Line 93.

**14. L117 : I my opinion equation 8 can be simplified, the second line should cover the case in the first line.**

**Author's Response:**

If  $tmpgrid_x = 0$ ,  $num_{x,tmp}$  obey the grid-distance nesting ratio and does not need to be adjusted. When use the second line in this situation, the  $num_x$  will be overestimated. That's why we defined two cases in Eq.8.

**15. L118 : int is the « integer part » function, please mention it**

**Author's Response:**

We replaced "int" by "integer" in Eq (9).

$$X_{min} = integer\left(\frac{xstart_{child} - xstart_{parent}}{dx_{par}}\right) \times dx_{par} + xstart_{parent} - dx_{child} * add_{w}$$

**16. L174 : what do the authors mean by « census » here ?**

**Author's Response:**

We modified this description on data. Please see "Provincial census of pollutant sources in transportation and residential sectors in Beijing (BMEEP, 2020) were allocated into 3km×3km grids by the "Mapinv" module." on Line 178.

**17. L180 : Please provide the meaning of GIS.**

**Author's Response:**

We replaced full name of GIS in this sentence. Please see "Shapefile formatted data can be displayed directly in ArcGIS or other geographic information system to display and check the configuration for nested domains" on Line 184.

18. L185-186 : the last sentence needs to be reformulated, it is difficult to understand the wording « demand »

**Author's Response:**

We explained the requirement of lateral boundary in AQM and rephased last sentence. Please see "Moreover, due to the outermost cells in WRF are inappropriate to use in AQM(USEPA,2019), "*model\_clip*" was conduct to define lateral boundary for AQM, as shown in **Appendix. A**. The major steps and key parameters are described below." on Line 86.

Please see "Due to blend of larger-scale driving data and scale-specific physics, the outermost cells in WRF domain are inappropriate to use and usually removed in AQM (USEPA,2019). The ISAT-configured domains reflect this requirement of lateral boundary in AQM and accurately obtain the WRF-AQM domain." on Line 189.

USEPA. 2019. CMAQ User's Guide. https://github.com/USEPA/CMAQ/blob/main/DOCS/

**Users\_Guide/**

**19. L188 : what is the resolution of MEIC here ?**

**Author's Response:**

We supplied the spatial resolution in MEIC. Please see "The MEIC regional emission inventory with spatial resolution of  $0.25^{\circ} \times 0.25^{\circ}$  in 2020 was downscaled to grid level in the "Downscale" module." on Line 179.

20. L189 : if I understand well, the authors compare MEIC downscaled by their routine with an independent BTH emission inventory ? right ? It is not clear is this section what the authors are comparing.

**Author's Response:**

Sorry for our unclear description in this section. Actually, we compared the gridded BTH emission inventory with different downscaled method, including intersect method, nearest method and sub-grid nearest method, as shown in Figure 6. We have rephased this section below.

"We compared gridded residential emissions in BTH under different downscaled method, including intersect method, nearest and sub-grid nearest methods." on Line 196.

21. L200 : what do the authors mean by «uncertainty in the boundary »?

**Author's Response:**

This statement may be inappropriate so we deleted it in the revision.

22. L207 : could the authors extend the analysis on other precursors emissions like NH3 that have a complete different spatial pattern ?

**Author's Response:**

Focus on developing an integrated tool on nested domain configuration and model-ready emission files in WRF-AQM. We haven't display emission characteristic of other pollutant.

I agree with the reviewers' opinion on the spatial pattern on different air pollutant. And we have published 10km\*10km gridded CH4 emission in China in 2020. We supplied a new cited reference in revision.

"Previous studies achieved reliable simulations result using this approach (H. Wang et al., 2021; K. Wang et al., 2022, 2021a, 2021b; Liu et al., 2023; Li, 2021; Tan, 2022)."

Liu S., Liu, K., Wang, K., et al. 2023. Fossil-Fuel and Food Systems Equally Dominate Anthropogenic Methane Emissions in China. Environmental Science & Technology. 2023, 57, 6, 2495–2505. https://doi.org/10.1021/acs.est.2c07933

23. L237 : what are « oceanfiles »?

**Author's Response:**

Ocean files are basic input in CMAQ model. We had supplied a new cited reference in this sentence.

"ISAT is highly extendable, including the creation of ocean files in CMAQ models (USEPA,2019)," on Line 250.

---

## Author Comment (AC2)

**Response to comments of the anonymous Referee #2, "RC2"**

**Comment 1**: This paper designs an integrated tool, ISAT v2.0, to configure nested model grid, downscale regional grided inventory, spatially allocate provincial/city-level inventory, and finally generate model-ready emission data. These are often the tedious and error-prone tasks that one has to do before running a WRF-AQM simulation, and ISAT v2.0 is designed to streamline this process with a clear and user-friendly workflow. In this sense, the modeling tool described in this paper definitely has its value and, if well implemented, can be very helpful to users. However, there are some tools that can accomplish similar tasks. Compared to those existing tools, I am not convinced that this new tool, as described in the current form, represents "substantial new concepts, ideas, or methods" (GMD criteria on scientific significance) in geoscientific modeling. A clear presentation of scientific novelty and contributions is required to meet the GMD standard.

**Author's Response:**

We appreciate the reviewer's insights. We agree with the reviewer that there are several public available platforms that can be used for processing emission inventory (e.g., Sparse Matrix Operator Kernel Emissions (SMOKE) developed by US EPA). However, these similar tools mainly focus on a single step in the WRF-AQM pre-processing. As described in introduction, "SMOKE, a Linux-platform supported and widely used tool in AQM, requires a predefined spatial surrogate from other geoprocessing tools, such as ArcGIS, **and cannot define parameters for nested domains in WRF-AQM** (Baek and Seppanen, 2021). The WRF Domain Wizard (https://esrl.noaa.gov/gsd/wrfportal/DomainWizard.html) allows the user to configure nested domains by **manually delimiting research areas.** However, **without the shapefile of the target area, obtaining precise domains is challenging, which requires several trials and expert experience to obtain suitable nested domains in AQMs**". Lacking an integrated and easy-to-use workflow, these are often the tedious and error-prone tasks that one has to do before running a WRF-AQM simulation, as you mentioned. This study integrates the steps in WRF-AQM preprocessing and conduct new algorithms in the ISAT v2.0. Benefit from integrated and easy-to-use workflow, the old version of ISAT has helped users to completed their research.

To clarify our scientific novelty and contributions to the modeling community, we rephrased the description of innovation and supplied related literature on ISAT's applications.

- Three references adopt ISAT had been supplied. Please see "Previous studies achieved reliable simulations result using this approach (H. Wang et al., 2021; K. Wang et al., 2022, 2021a, 2021b; Liu et al., 2023; Li, 2021; Tan, 2022)" on Line 235.
- Li, Y. 2021. Study on Ozone Formation Sensitivity in the Pearl River Delta based on Satellite Remote Sensing and Air qualify Model. Master's Thesis, South China University of Technology.
- Liu S., Liu, K., Wang, K., et al. 2023. Fossil-Fuel and Food Systems Equally Dominate Anthropogenic Methane Emissions in China. Environmental Science & Technology. 2023, 57, 6, 2495–2505. https://doi.org/10.1021/acs.est.2c07933
- Tan, X. 2022.Construction of CMAQ Pollution Source Inventory Based on ISAT Model. Master's Thesis, Jilin University.
- (2) Clarify the innovation of "Prepgrid" model in "2.1 Nested domain configuration in "Prepgrid"". In this module, we conduct a shapefile-based algorithm on nested domain configuration. And user can obtain parameters of nested domain including projection parameters, grid position and extent based on shapefiles in each domain. Please see "In practice, we usually obtain the extent of the study area based on its shapefile. Compared with manual configuration, using shapefile can provide consistent and accurate nested domain between WRF, AQM and emission inventory." on Line 87.
- (3) Previous emission processing tool, such as SMOKE, allocate local emission inventory based on processed spatial surrogate by ArcGIS and cannot downscaled regional emission inventory. In ISAT, we integrated and simply these steps in "Mapinv" and "Downscale" without external tools such as ArcGIS. And sub-grid nearest method was conduct in

"Downscale", which obtained downscale emission inventory user-friendly and easy-to-use. Please see "This module can downscale regional emission inventories user-friendly and easy-to-use based on default or user-defined proxy without external tools such as ArcGIS." on Line 136.

(4) SMOKE model applied Source Classification Code (SCC), GSCNV, GSPRO, GSPRODESC, GSPRO\_COMBO, GSREF, and GSTAG files to produce model-ready emission inventory in CMAQ. ISAT adopted a simpler and user-friendly source classification and speciation method, and allows users to modify chemical mechanism in different AQM according to their needs. We supplied the innovation of prepmodel module. Please see "Users can easily add or delete sources in model-ready emission inventory by their needs in "Prepmodel"." on Line 154.

**Comment 2**: The authors present "sub-grid nearest" method as an innovation. This is not a new idea either. In section 3.2, the "sub-grid nearest" method is compared with the nearest method and the intersect method. The accuracy of the method is shown with R2, but the claimed computing efficiency of the "sub-grid nearest" method is not discussed.

**Author's Response:**

Thanks for your comment. Traditional nearest method ignored the differences of resolution between regional emission inventory and targe domain in WRF-AQM. Conducting sub-grid ratio, sub-grid nearest method can optimize the results on nearest method and obtain more accurate results. Meanwhile, we supplied the discussion of computing efficiency. Pleases see "Compared with ArcGIS, SMOKE and other tools, using "Dowanscale" module reduces the timeliness from hours to minutes." on Line 195. And "For example, the running time of "downscaled" module increased from 2 minutes to 10 minutes with the sub-grid ratio increased from 3 to 9 in this case." on Line 204.